# TESGNN: Temporal Equivariant Scene Graph Neural Networks for Efficient and Robust Multi-View 3D Scene Understanding

**Quang P. M. Pham**[+]                                    *quang.pham@mbzuai.ac.ae*
*Department of Robotics*
*Mohamed bin Zayed University of Artificial Intelligence, UAE*

**Khoi T. N. Nguyen**[+]                          *nguyentietnguyenkhoi@gmail.com*
*College of Engineering and Computer Science*
*VinUniversity, Vietnam*

**Lan C. Ngo**                                          *20lan.nc@vinuni.edu.vn*
*College of Engineering and Computer Science*
*VinUniversity, Vietnam*

**Truong Do**                                          *truong.dt@vinuni.edu.vn*
*College of Engineering and Computer Science*
*VinUniversity, Vietnam*

**Dezhen Song**                                       *dezhen.song@mbzuai.ac.ae*
*Department of Robotics*
*Mohamed bin Zayed University of Artificial Intelligence, UAE*

**Truong Son Hy**[*]                                            *thy@uab.edu*
*Department of Computer Science*
*The University of Alabama at Birmingham, US* [*]

**Reviewed on OpenReview:** *https://openreview.net/forum?id=boMOkkYPzE*

## Abstract

Scene graphs have proven to be highly effective for various scene understanding tasks due to their compact and explicit representation of relational information. However, current methods often overlook the critical importance of preserving symmetry when generating scene graphs from 3D point clouds, which can lead to reduced accuracy and robustness, particularly when dealing with noisy, multi-view data. Furthermore, a major limitation of prior approaches is the lack of temporal modeling to capture time-dependent relationships among dynamically evolving entities in a scene. To address these challenges, we propose Temporal Equivariant Scene Graph Neural Network (TESGNN), consisting of two key components: (1) an Equivariant Scene Graph Neural Network (ESGNN), which extracts information from 3D point clouds to generate scene graph while preserving crucial symmetry properties, and (2) a Temporal Graph Matching Network, which fuses scene graphs generated by ESGNN across multiple time sequences into a unified global representation using an approximate graph-matching algorithm. Our combined architecture TESGNN shown to be effective compared to existing methods in scene graph generation, achieving higher accuracy and faster training convergence. Moreover, we show that leveraging the symmetry-preserving property produces a more stable and accurate global scene representation compared to existing approaches. Finally, it is computationally efficient and easily implementable using existing

---

[*]Corresponding author. [+]Equal contribution.

frameworks, making it well-suited for real-time applications in robotics and computer vision. This approach paves the way for more robust and scalable solutions to complex multi-view scene understanding challenges.

# 1 Introduction

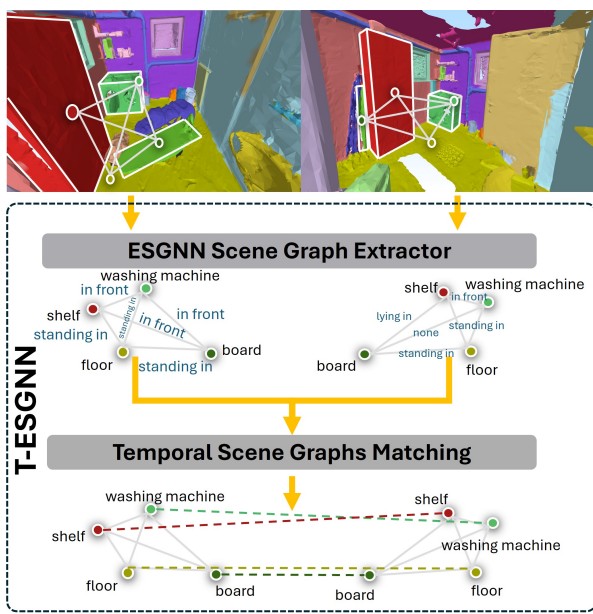

Figure 1: A visualization of a multi-view scene graph from multiple 3D point cloud sequences. Our proposed TESGNN first generates local scene graphs for each sequence using Equivariant GNN. Then, the local scene graphs are merged by passing through a temporal layer to form a global scene graph representing the entire scene.

Holistic scene understanding plays a critical role in the advancement of robotics and computer vision systems Kim et al. (2024); Li et al. (2022b); Koch et al. (2024a). The primary methodologies in this domain include 2D mapping, 3D reconstruction, and scene graph representation. Although 2D mapping remains a straightforward and widely adopted approach Pfaff et al. (2006), it inherently lacks a comprehensive spatial understanding of the environment. In contrast, 3D reconstruction techniques Jin et al. (2024) provide richer spatial information but are resource intensive and particularly susceptible to noise interference Li et al. (2022a). Recent innovations Shah et al. (2022); Huang et al. (2023); Yamazaki et al. (2024) leverage Vision Language Models (VLMs) and Large Language Models (LLMs) to generate semantic maps from 3D data, thereby enhancing scene understanding. However, these approaches often require substantial computational resources. As an alternative, scene graphs, particularly those utilizing Graph Neural Networks (GNNs), offer a more efficient and semantically rich representation compared to traditional 3D reconstruction techniques Chang et al. (2023). Initial methodologies mainly focused on generating scene graphs from sequences of 2D images. In more recent work Wald et al. (2020); Wu et al. (2021b), researchers have explored scene graph generation by incorporating 3D features, such as depth data and point clouds. Latest advancements Wu et al. (2023); Koch et al. (2024b) propose hybrid, multi-modal approaches that integrate both 2D and 3D data to further refine scene graph representations. These hybrid approaches often yield better prediction results by combining image encoders or VLMs with GNNs for scene graph generation.

However, based on our analysis, existing GNN architectures for scene graph generation have several limitations. Current methods struggle with noisy, multi-view 3D data, leading to inconsistent scene graphs due to sensitivity to camera angles and poor symmetry preservation. Enhancing GNNs with invariant node and edge embeddings and using Equivariant GNNs Köhler et al. (2020); Liao & Smidt (2023) can address this, though prior works Wu et al. (2021b; 2023) haven't explored these approaches. Additionally, existing

methods lack robust strategies for building coherent global scene graphs from multiple point cloud sequences. For example, Wu et al. (2021b) uses heuristic averaging to merge local graphs but skips optimization, which could potentially result in global graphs that lack semantic coherence and structural consistency.

We introduce the **Temporal Equivariant Scene Graph Neural Network (TESGNN)**, a novel architecture that combines two key innovations:

- **Equivariant Scene Graph Neural Network (ESGNN) for Scene Graph Extraction**: extracts scene graphs from 3D point clouds using an Equivariant GNN with Equivariant Graph Convolution and Feature-wise Attention layers, ensuring rotational and translational invariance. This design delivers high-quality scene representations with superior accuracy, fewer training iterations, and reduced computational demands compared to existing methods - marking the first application of symmetry-preserving Equivariant GNNs for this task.

- **Temporal Scene Graph Matching for Global Scene Representation**: fuses the local scene graphs into a unified global representation by solving a graph matching problem based on node embedding similarity, with ESGNN as the backbone scene graph extractor. This embedding-based approach is robust for multi-view scene understanding, being more effective compared to other methods and enabling applications in multi-agent coordination and navigation in dynamic environments.

## 2 Related Work

**Graph Representation and Equivariance** Graph representation learning encodes structural and relational data into low-dimensional embeddings using methods like GNNs, Graph Transformers Yun et al. (2019); Kim et al. (2022); Cai et al. (2023), and Graph Attention Networks Veličković et al. (2018), enabling tasks such as scene graph understanding without dense 3D data. For 3D applications, standard GNNs lack rotational symmetry; recent SE(3)/E(3)-equivariant architectures like EGNN Satorras et al. (2021) and Equiformer Liao & Smidt (2023) address this by preserving geometric symmetries during transformations, critical for point clouds Uy et al. (2019), molecular structures, and spatial reasoning.

**Scene Graph Understanding** Research on scene graphs has significantly advanced tasks in vision, natural language processing, and interdisciplinary domains such as robot navigation Honerkamp et al. (2024); Pham et al. (2025). Originally introduced as a structure to represent object instances and their relationships within a scene Johnson et al. (2015), scene graphs effectively capture rich semantics in various data modalities, including 2D/3D images Johnson et al. (2015); Armeni et al. (2019) and videos Qi et al. (2018). 3DSSG Wald et al. (2020) pioneered scene graph generation from 3D point clouds. Building on this, SGFN Wu et al. (2021b) employs Feature-wise Attention to improve scene graph representations, followed by an incremental approach Wu et al. (2023) that integrates both RGB image sequences and sparse 3D point clouds. To the best of our knowledge, SGFN remains state-of-the-art for scene graph generation from 3D point clouds, without incorporating image encoder or VLMs. Our approach adopts this Feature-wise Attention to optimize the Message-Passing process and enhance the generated scene graphs. Additionally, it is important to note that while recent works Wu et al. (2023); Koch et al. (2024b); Saxena et al. (2025) focus on utilizing vision–language encoders, our work investigates a different problem: How to improve the GNN backbone design for 3D understanding, leading to better generalization and faster convergence.

**Temporal Graph Learning** Temporal graph learning has gained significant attention for modeling dynamic relationships over time, with applications in traffic prediction Yu et al. (2018); Li et al. (2018); Nguyen et al. (2024a). Traditional GNNs assume static graph structures, limiting their applicability in scenarios like video analysis and multi-view scene understanding. To address this, Temporal GNNs Rossi et al. (2020; 2023) incorporate time-series learning to capture evolving patterns. In scene graph matching, methods such as SGAligner Sarkar et al. (2023) and SG-PGM Xie et al. (2024) align nodes across graphs but rely on ground truth graphs, making them impractical for real-world tasks where predicted graphs contain noise, ambiguous edges, and node permutations. To overcome these challenges, we introduce a symmetry-preserving temporal layer in ESGNN, leveraging equivariant properties to merge scene graphs across time steps. Unlike prior approaches that treat temporal information as sequential snapshots, our method integrates graph matching to construct a unified representation.

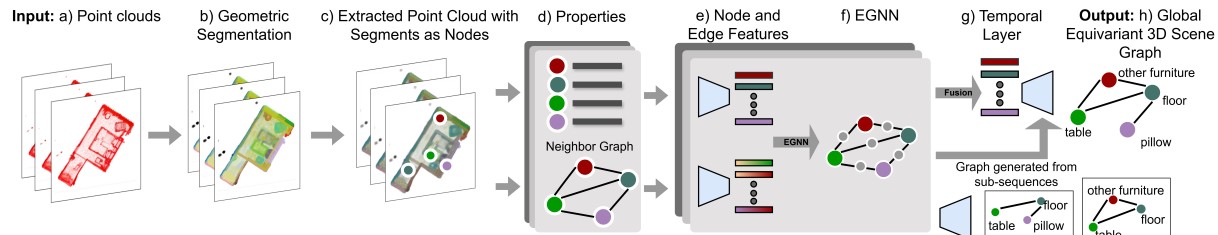

Figure 2: Overview of our proposed TESGNN. Our approach takes sequences of point clouds a) as input to generate a geometric segmentation b). Subsequently, the properties of each segment and a neighbor graph between segments are constructed. The properties d) and neighbor graph e) of the segments that have been updated in the current frame c) are used as the inputs to compute node and edge features f) and to predict a 3D scene graph g). Then it goes through the temporal layer to fuse graphs from different sequences to a global one h).

**Large-scale Point Cloud Reconstruction**  Reconstructing large-scale point clouds is inherently challenging due to their irregular structure, often comprising multiple sub-point cloud sequences Huang et al. (2022). To tackle this issue, we introduce a temporal model that merges graphs generated from these sequences by utilizing embeddings from our proposed models. This approach not only improves integration across sequences but also extends to applications such as multi-agent SLAM and dynamic environment adaptation Jiang et al. (2019); Zou et al. (2019), offering an efficient, lightweight solution based on multi-view scene graph representations.

## 3  Overall Framework

Fig. 2 illustrates the proposed framework's capability to iteratively estimate a global 3D semantic scene graph from a sequence of point clouds. The framework consists of three key phases: Feature Extraction (a-c), discussed in Section 3.2; Scene Graph Extraction (d-f); and Temporal Graph Matching (g-h), detailed in Section 3.3. Our main works, scene graph extraction and temporal graph matching, are further elaborated in Sections 4 and 5.

### 3.1  Problem Formulation

Our system processes point cloud representations $D_i$ for each scene $i$ in a sequence $(D_1, D_2, \ldots, D_n)$ to generate corresponding scene graphs $(\mathcal{G}_{s,1}, \mathcal{G}_{s,2}, \ldots, \mathcal{G}_{s,n})$ and a global graph $\mathcal{G}_{s,\text{global}}$ that aggregates scene information. Assuming effective feature extraction and geometric segmentation, our focus is on scene graph generation rather than feature extraction. Each point cloud $D_i$ undergoes pre-processing for input to a graph neural network (GNN). A geometric segmentation model partitions $D_i$ into segments, forming nodes of the scene graph, with segment attributes extracted via a point encoder. Detailed feature extraction methods are in Section 3.2.

We model symmetry preservation using the Euclidean group $E(3)$, which encompasses 3D rotations and translations. By employing equivariant layers, our model remains robust to variations in object orientation and position, enhancing generalization and performance while accelerating convergence in scene graph generation.

The Semantic Scene Graph is denoted as $\mathcal{G}_s = (\mathcal{V}, \mathcal{E})$, where $\mathcal{V}$ and $\mathcal{E}$ represent sets of entity nodes and directed edges, respectively. In this case, each node $v_i \in \mathcal{V}$ contains an entity label $l_i \in L$, point clouds $\mathcal{P}_i$, an 3D Oriented Bounding Box (OBB) $b_i$, and a node category $c_i^{\text{node}} \in \mathcal{C}^{\text{node}}$. Conversely, each edge $e_{i \to j} \in \mathcal{E}$, connecting node $v_i$ to $v_j$ where $i \neq j$, is characterized by an edge category or semantic relationship denoted by $c_{i \to j}^{\text{edge}} \in \mathcal{C}^{\text{edge}}$, or can be written in a relation triplet ⟨*subject, predicate, object*⟩. Here, $L$, $\mathcal{C}^{\text{node}}$, and $\mathcal{C}^{\text{edge}}$ represent the sets of all entity labels, node categories, and edge categories, respectively. Given the 3D scene data $D_i$ and $D_j$ that represent the same point cloud of a scene but from different views (rotation and transition), we try to predict the probability distribution of the equivariant scene graph prediction in which the equivariance is preserved:

$$\begin{cases} P(\mathcal{G}|D_i) = P(\mathcal{G}|D_j)_{i \neq j} \\ D_j = R_{i \to j}D_i + T_{i \to j} \end{cases} \tag{1}$$

where $R_{i \to j}$ is the rotation matrix and $T_{i \to j}$ is the transition matrix.

We denote a global point cloud, that is,

$$D_{\text{global}} = \Phi_{\text{matching}}\left(D_1, D_2, ..., D_n\right),$$

where $\Phi_{\text{matching}}$ is a model / algorithm to align subsequences together, we can define the global scene graph distribution as $P(\mathcal{G}_{s,\text{global}}|D_{\text{global}})$. However, $\Phi_{\text{matching}}$ is hard to define, especially in the case that the origin coordinate of each sequence is not aligned, which consumes time and resources for sampling or large estimation models. Instead, as the symmetry-perserving property is maintained, the embedding between each graph is similar, it is much easier to perform the matching between the graph embedding vectors, so that:

$$\begin{aligned} P(\mathcal{G}_{s,\text{global}}|D_{\text{global}}) &= P(\mathcal{G}_s|\Phi_{\text{matching}}((D_1, D_2, ..., D_n)) \\ &= \Phi_{\text{graph}}\left(\mathcal{G}_{s,1}, \mathcal{G}_{s,2}, ..., \mathcal{G}_{s,n}\right) \end{aligned}$$

where $\Phi_{graph}$ is the model matching the embedding vectors together, which will be described in a later section.

### 3.2 Feature Extraction

In this phase, the framework extracts the features for scene graph generation, following the two main steps:

1. **Point Cloud Reconstruction:** The proposed framework takes the point cloud data, which can be reconstructed from various techniques such as ORB-SLAM3 or HybVIO Campos et al. (2021) as the input. However, for the objective validation purpose of scene graph generation, we use the indoor point cloud dataset 3RScan Wald et al. (2019) for ground truth data $D_i$.

2. **Geometric Segmentation and Point Cloud Extraction with Segment Nodes:** Given a point cloud $D_i$, this geometric segmentation will provide a segment set $\mathbf{S} = \{\mathbf{s}_1, ..., \mathbf{s}_n\}$. Each segment $s_i$ consists of a set of 3D points $\mathbf{P}_i$ where each point is defined as a 3D coordinate $p_i \in \mathbb{R}^3$ and RGB color. Then, the point cloud concerning each entity is fed to the point encoder named PointNet Charles et al. (2017) $f_p(\mathbf{P_i})$ to encode the segments $s_i$ into latent node and edge features, which are then passed to the model detailed in Section 4.

### 3.3 Scene Graph Extraction and Temporal Graph Matching

In this phase, the framework processes the input from feature extraction (Section 3.2) to generate the scene graph:

- **Properties and Neighbor Graph Extraction:** From the point cloud, we extract features including the centroid $\overline{\mathbf{p}}_i \in \mathbb{R}^3$, standard deviation $\boldsymbol{\sigma}_i \in \mathbb{R}^3$, bounding box size $\mathbf{b}_i \in \mathbb{R}^3$, maximum length $l_i \in \mathbb{R}$, and volume $\nu_i \in \mathbb{R}$. We create edges between nodes only if their bounding boxes are within 0.5 meters of each other, following Wu et al. (2021b).

- **Scene Graph Extraction and Temporal Graph Matching:** The processed input from above is then fed to the ESGNN (Section 4) to generate the node and edge embeddings. These embeddings are used to generate the sub-graph for each sequence and merge multiple scene graphs with our Temporal Model (Section 5). For ESGNN, the node classes and edge predicates are predicted using two Multi-Layer Perceptron (MLP) classifiers. ESGNN is trained end-to-end with a joint cross-entropy loss for both objects $\mathcal{L}_{\text{obj}}$ and predicates $\mathcal{L}_{\text{pred}}$, as described in Wald et al. (2020). Meanwhile, our Temporal Graph Matching is trained to minimize the representation distance between identical nodes with contrastive loss Hadsell et al. (2006).

# 4 Equivariant Scene Graph Neural Network (ESGNN)

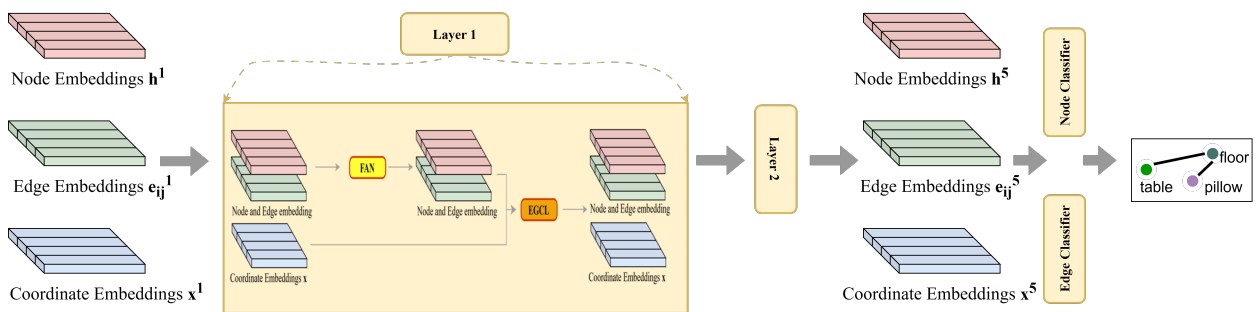

Figure 3: ESGNN Scene Graph Extractor pipeline. The model comprises two main layers: (1) Feature-wise Attention Graph Convolution Layer (FAN-GCL) and (2) Equivariant Graph Convolution Layer (EGCL). FAN-GCL handles large inputs with multi-head attention to update node and edge features, while EGCL ensures symmetry preservation by incorporating bounding box coordinates into the message-passing mechanism. ESGNN leverages these layers to maintain rotation and permutation equivariance, thus enhance the quality of scene graph generation.

To build a network architecture that effectively generate scene graph from point cloud, we propose a combination of Equivariant Graph Convolution Layers Satorras et al. (2021) and the Graph Convolution Layers with Feature-wise Attention Wu et al. (2021b).

## 4.1 Graph Initialization

**Node features**  The node feature includes the invariant features $\mathbf{h_i}$ and the 3-vector coordinate $\mathbf{x_i} \in \mathbb{R}^3$. The invariant features $\mathbf{h}_i$ consists of the latent feature of the point cloud after going through the PointNet $f_p(\mathbf{P}_i)$, standard deviation $\boldsymbol{\sigma}_i$, log of the bounding box size $\ln(\mathbf{b}_i)$ where $\mathbf{b}_i = (b_x, b_y, b_z) \in \mathbb{R}^3$, log of the bounding box volume $\ln(v_i)$ where $v_i = b_x b_y b_z$, and log of the maximum length of bounding box $\ln(l_i)$. The coordinate of the bounding box $x_i$ is defined by the coordinates of the two furthest corners of the bound box. The formula can be written as follows:

$$\mathbf{v}_i = (\mathbf{h}_i, \mathbf{x}_i),$$
$$\mathbf{h}_i = [f_p(\mathbf{P}_i), \boldsymbol{\sigma}_i, \ln(\mathbf{b}_i), \ln(\nu_i), \ln(l_i)],$$
$$\mathbf{x}_i = [\mathbf{x}_i^{\text{bottom right}}, \mathbf{x}_i^{\text{top left}}].$$

**Edge features**  The visual characteristics of the edges are determined by the properties of the connected segments. For an edge between a source node $i$ and a target node $j$ where $j \neq i$, the edge visual feature $\mathbf{e}_{ij}$ is computed as follows:

$$\mathbf{r}_{ij} = \left[\overline{\mathbf{p}}_i - \overline{\mathbf{p}}_j, \boldsymbol{\sigma}_i - \boldsymbol{\sigma}_j, \mathbf{b}_i - \mathbf{b}_j, \ln\left(\frac{l_i}{l_j}\right), \ln\left(\frac{\nu_i}{\nu_j}\right)\right],$$
$$\mathbf{e}_{ij} = g_s(\mathbf{r}_{ij}),$$

where $g_s(\cdot)$ is a Multi-Layer Perceptron (MLP) that projects the paired segment properties into a latent space.

## 4.2 Graph Neural Network Architecture and Weights Updating

Fig. 3 describes our GNN network with two core components: ① Feature-wise Attention Graph Convolution Layer (FAN-GCL); and ② Equivariant Graph Convolution Layer (EGCL). The FAN-GCL, proposed by Wu et al. (2021b), is used to handle the large input queries $Q$ of dimensions $d_q$ and targets $T$ of dimensions $d_\tau$ by utilizing multi-head attention. On the other hand, EGCL, proposed by Satorras et al. (2021), is used to

maintain symmetry-preserving equivariance, allowing us to incorporate the bounding box coordinates $x_i$ as node features and update them through the message-passing mechanism.

ESGNN has 2 main layers, each consisting of a FAN-GCL followed by an EGCL, forming a total of 4 message-passing layers. For each main layer, the formulas for updating node and edge features are defined as:

- Update FAN-GCL:

$$\mathbf{v}_i^{\ell+1} = g_v \left( \left[ \mathbf{v}_i^\ell, \max_{j \in \mathcal{N}(i)} \left( \text{FAN} \left( \mathbf{v}_i^\ell, \mathbf{e}_{ij}^\ell, \mathbf{v}_j^\ell \right) \right) \right] \right),$$
$$\mathbf{e}_{ij}^{\ell+1} = g_e \left( \left[ \mathbf{v}_i^\ell, \mathbf{e}_{ij}^\ell, \mathbf{v}_j^\ell \right] \right),$$

- Update EGCL:

$$h_i^{(l+1)} = h_i^{(l)} + \mathrm{g_v} \left( \text{concat} \left( h_i^{(l)}, \sum_{j \in \mathcal{N}(i)} e_{ij}^{(l)} \right) \right),$$
$$e_{ij}^{(l+1)} = \mathrm{g_e} \left( \text{concat} \left( h_i^{(l)}, h_j^{(l)}, \|\mathbf{x}_i^{(l)} - \mathbf{x}_j^{(l)}\|^2, e_{ij}^{(l)} \right) \right),$$
$$\mathbf{x}_i^{(l+1)} = \mathbf{x}_i^{(l)} + \sum_{j \in \mathcal{N}(i)} (\mathbf{x}_i^{(l)} - \mathbf{x}_j^{(l)}) \cdot \phi_{\text{coord}}(e_{ij}^{(l)}).$$

**Equivariance of ESGNN**    We provide a detailed proof of equivariance in the Appendix B.

## 5    Temporal Graph Matching Network

Leveraging the symmetry-preserving properties of our scene graph extractor, we hypothesize that its node and edge embeddings can remain inherently distinguishable, especially for *isomorphic* scene graphs subjected to rotations and translations. This unique feature opens the gate for efficient temporal scene graph matching: Given multiple point cloud sequences containing overlapping regions, our model can reliably match the corresponding parts, regardless of viewpoint changes. As a result, multiple scene graphs can be aligned and fused into a unified, spatially consistent global graph over time.

Figure 4 illustrates our graph matching model. The core idea is to generate a compact representation for each unique triplet ⟨*subject, predicate, object*⟩, then merge identical ones by their similarity. This approach avoids explicit graph isomorphism search, while being designed to ensure order-invariant and scale to graphs of any size.

**Triplet Representation** For each triplet ⟨*subject, predicate, object*⟩ we:

- Concatenate the predicate (edge) embedding with the Object-node embedding to produce a Predicate–Object vector.

- Pass the Predicate–Object vectors connected to the same Subject through a linear layer and sum-pooled to ensure permutation invariance.

- Fuse the pooled vector with the Subject-node embedding, then refine it through a second linear layer and a self-attention block to capture higher-order context.

The result is a compact, rotation-invariant Triplet embedding for every local scene graph.

**Similarity-based Graph Matching** Given two sequences, we compute cosine similarities between all Triplet embeddings and perform top-K retrieval. Pairs whose similarity exceeds a fixed threshold are treated as the same physical object; the corresponding local graphs are merged node by node to build the global scene graph.

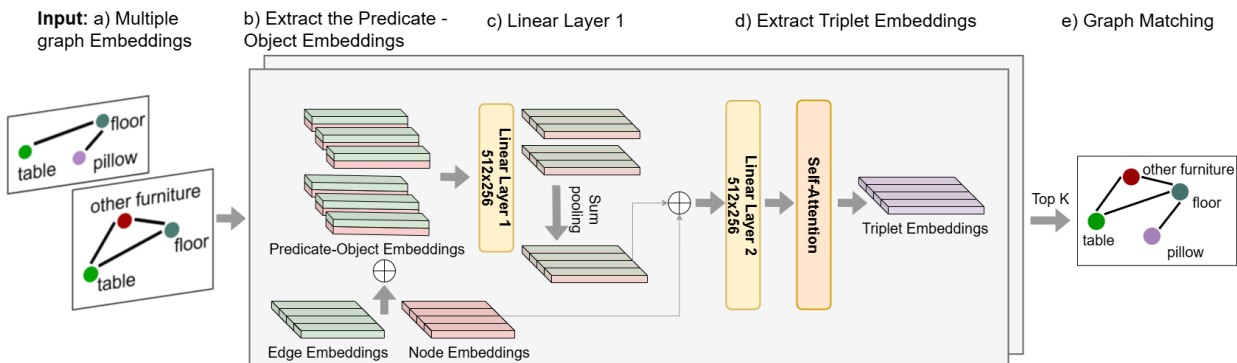

Figure 4: Temporal Graph Matching pipeline. First, node and edge embeddings derived from scene graphs of different sequences (a). For each sequence, edge embeddings are concatenated with the target node embeddings to create Predicate-Object embeddings (b), which then pass through a linear layer followed by sum pooling (c). For each node, the embeddings from all associated edges are concatenated and processed through another linear layer and self-attention mechanism to generate the final representation of each segment (d). These final Triplet Embeddings are utilized for top-K retrieval graph matching (e).

**Training Objective** We train the graph matching model such that distances between similar Triplet embeddings are minimized, while those between dissimilar embeddings are maximized. We achieve this using Contrastive Loss Hadsell et al. (2006), a well-established method for training embeddings Reimers & Gurevych (2019). Unlike traditional loss functions that aggregate over samples, Contrastive Loss operates on pairs:

$$\mathcal{L}(W, Y, X_1, X_2) = \frac{1 - Y}{2}(D_W)^2 + \frac{Y}{2} \max(0, m - D_W)^2,$$

where $(Y, X_1, X_2)$ are pairs of Triplet Embeddings, $D_W$ is a distance function, and $m > 0$ is a margin. We use Siamese Cosine Similarity Reimers & Gurevych (2019) for $D_W$ with hard-positive / hard-negative mining to focus learning on the most ambiguous cases.

## 6 Experiment

We evaluate TESGNN on the 3DSSG dataset Wald et al. (2020) and compare the results with state-of-the-art works. Section 6.3 provides results for scene graph generation from 3D point clouds, in comparison to 3DSSG Wald et al. (2020) and Scene Graph Fusion (SGFN) Wu et al. (2021b). Our method is evaluated on full scenes given geometric segments mentioned in Section 3.2. Section 6.5 reports our Temporal Graph Matching.

### 6.1 Dataset

**Scene Graph Extraction:** We use the 3DSSG Wald et al. (2020) [1] - a popular dataset for 3D scene graph evaluation built upon 3RScan Wald et al. (2019), adapting the setting from SGFN Wu et al. (2021b). At the time this paper is written, 3DSSG is the only dataset for semantic scene graph generation. The original 3RScan contains 1482 3D reconstructions/snapshots of 478 naturally changing indoor environments. After being processed with ScanNet Dai et al. (2017) for geometric segmentation and ground truth scene graph generation, the final dataset consists of 1061 sequences from 382 scenes for training, 157 sequences from 47 scenes for validation, and 117 sequences from 102 scenes for testing. We tested TESGNN on both 2 dataset versions **l20** and **l160**. The **l20** contains 20 objects and 8 predicates, and **l160** contains 160 objects and 26 predicates.

**Temporal Graph Matching:** While reproducing prior works including SGAligner Sarkar et al. (2023) and SG-PGM Xie et al. (2024), we identified limitations in their dataset. Since their work does not contain

---
[1]https://github.com/ShunChengWu/3DSSG

scene graph extraction, they synthesize scene graph node and edge features using one-hot encoding of the ground truth labels, which does not accurately reflect scene graph embeddings. This setup impractically ignores real-world noises affecting local scene graphs. To address these shortcomings, we leverage our above setup for scene graph extraction to evaluate scene graph matching, using the same sequences and scenes. In this context, a positive pair consists of the same object appearing in two or more different sequences. There are 16,156 positive pairs over 321,292 total pairs for the train set, and 3,062 positive pairs over 63,890 total pairs for the test set. In our setup, node and edge features are derived from frozen scene graphs such as ESGNN (ours) or SGFN Wu et al. (2021b).

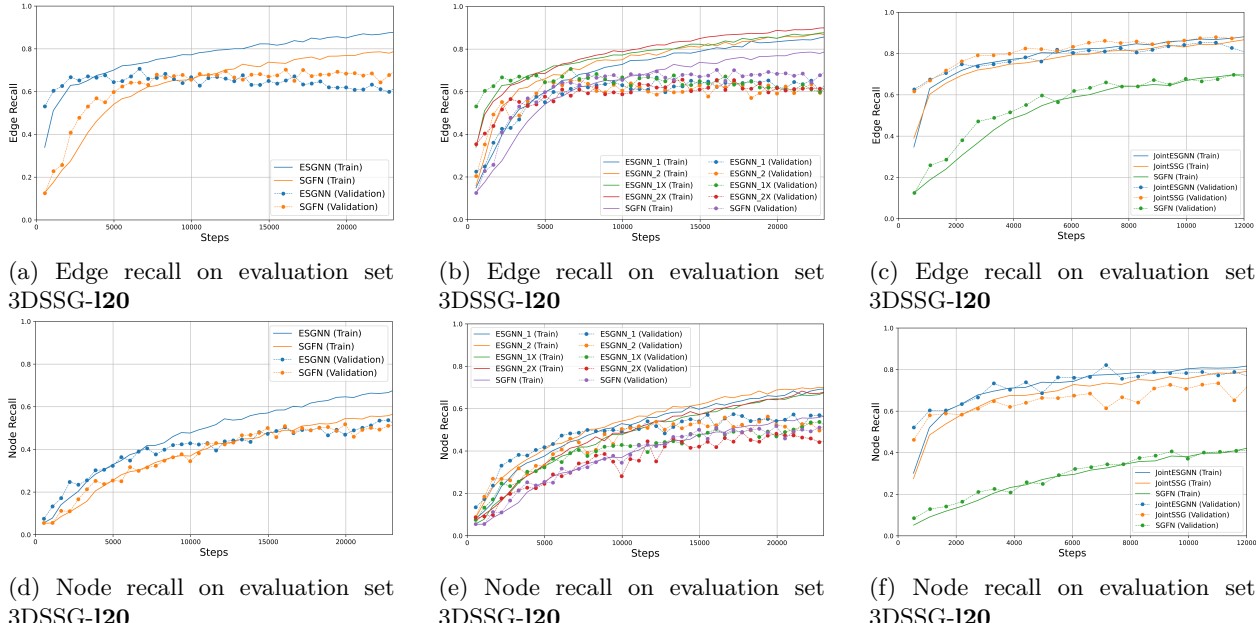

(a) Edge recall on evaluation set 3DSSG-**l20**

(b) Edge recall on evaluation set 3DSSG-**l20**

(c) Edge recall on evaluation set 3DSSG-**l20**

(d) Node recall on evaluation set 3DSSG-**l20**

(e) Node recall on evaluation set 3DSSG-**l20**

(f) Node recall on evaluation set 3DSSG-**l20**

Figure 5: Comparison of recall result over training steps, column-wise interpretation. (a), (d) illustrates the result of our model versus SGFN. (b), (e) shows the ablation study. (c), (f) shows the result while applying our model with the image encoder.

## 6.2 Metrics

**Scene Graph Extraction:** We mainly use the **Recall** of node and edge as our evaluation metrics, given that the dataset is unbalanced Wald et al. (2020) and the objective of scene graphs is to effectively capture the semantic meaning of the surrounding world. In the training phase, we calculate the **Recall** as the true positive overall positive prediction. In the test phase, for more detailed analysis, we use **Recall@k (R@k)** Wald et al. (2020); Wu et al. (2021b; 2023), which takes **k** most confident predictions and marks them as correct if at least one of these predictions is correct. For a relationship triplet, **R@1** is the accuracy of the predictions. We apply the recall metrics for the predicate (edge classification), object (node classification), and relationship (triplet ⟨*subject, predicate, object*⟩). Additionally, we report the **Number of Params** and **Runtime** for inference.

**Temporal Graph Matching:** We follow the Information Retrieval evaluation metrics, including **R@k**, **MRR@k**, and **mAP@k** [2]. The **R@k** used in this evaluation is slightly different: It is the proportion of instances where at least one correct match appears among the top-*k* matches with a **confidence score** above 0.5. If the matching score is below 0.5, the node is considered unique, indicating that it does not have a common match across sequences. **MRR@k** evaluates the effectiveness of retrieval by averaging the reciprocal ranks of the *k* similar nodes. **mAP@k** measures the quality of rankings by emphasizing *k* relevant nodes at higher ranks.

---

[2]Since each node has only one ground-truth match, mAP@k and MRR@k produce similar result in our setting.

### 6.3 Scene Graph Extraction Evaluation

| Method | Relationship | | Object | | Predicate | | Recall | |
|--------|------|------|------|------|------|------|------|------|
| | R@1 | R@3 | R@1 | R@3 | R@1 | R@2 | Obj. | Rel. |
| 3DSSG | 32.65 | 50.56 | 55.74 | 83.89 | **95.22** | 98.29 | 55.74 | **95.22** |
| SGFN | 37.82 | 48.74 | 62.82 | **88.08** | 81.41 | 98.22 | 63.98 | 94.24 |
| Ours | **43.54** | **53.64** | **63.94** | 86.65 | 94.62 | **98.30** | **65.45** | 94.62 |

(a) **3DSSG-l20**

| Method | Relationship | | | Object | | | Predicate | | | Recall | |
|--------|------|------|------|------|------|------|------|------|------|------|------|
| | R@1 | R@5 | R@10 | R@1 | R@5 | R@10 | R@1 | R@5 | R@10 | Obj. | Rel. |
| 3DSSG | 64.80 | 68.69 | 69.91 | 27.33 | 60.82 | 73.59 | **67.25** | **96.72** | **98.63** | 27.33 | **21.65** |
| SGFN | 64.69 | 68.22 | 69.37 | 35.76 | 67.55 | **79.40** | 48.63 | 92.33 | 97.71 | 35.76 | 14.09 |
| Ours | **65.00** | **69.38** | **70.46** | **37.46** | **67.97** | 79.14 | 36.88 | 90.14 | 96.89 | **37.46** | 20.91 |

(b) **3DSSG-l160**

Table 1: Scene graph prediction performance on l20 and l160 datasets, including relationship triplets, objects, and predicates. *Recall* columns reporting object and relationship recall scores (Obj., Rel.). As the l160 dataset contains more objects compared to l20, we provide a wider range of **R@k** for better assessment.

**Effective prediction outcome:** Tables 1a and 1b compare the results between ESGNN (ours) with existing models 3DSSG Wald et al. (2020) and SGFN Wu et al. (2021b) with a geometric segmentation setting. Our method obtains high results in both relationship, object, and predicate classification. Especially, ESGNN outperforms the existing methods in relationship prediction and obtains significantly higher **R@k** in predicate compared to SGFN. For the l160 dataset, while TESGNN maintains consistently stronger performance in relationship and object prediction, we observe a decrease in predicate prediction compared to existing methods. We attribute this to the influence of equivariance when distinguishing directional relationships (e.g., "left" vs. "right"), which appear more frequently in the l160 than in l20. Additionally, the authors of SGFN and 3DSSG suggested that the higher predicate recall in 3DSSG may come from the use of union 3D bounding boxes, which could be more suitable for estimating multiple predicates Wu et al. (2021a). Nevertheless, we highlight the high accuracy in relationship triplet prediction of our approach. This is crucial, as it demonstrates that ESGNN is effective in capturing semantic relationships between objects, a key factor in generating accurate scene graphs. For predictions on unseen data, ESGNN performs competitively with SGFN, as shown in Tables 2a and 2b.

| Method | New Relationship | | New Object | | New Predicate | |
|--------|------|------|------|------|------|------|
| | R@1 | R@3 | R@1 | R@3 | R@1 | R@2 |
| 3DSSG | 39.74 | 49.79 | 55.89 | 84.42 | **70.87** | 83.29 |
| SGFN | **47.01** | 55.30 | 64.50 | **88.92** | 68.71 | **83.76** |
| Ours | 46.85 | **56.95** | **65.47** | 87.52 | 66.90 | 82.88 |

(a) **3DSSG-l20**

| Method | New Relationship | | | New Object | | | New Predicate | | |
|--------|------|------|------|------|------|------|------|------|------|
| | R@1 | R@5 | R@10 | R@1 | R@5 | R@10 | R@1 | R@5 | R@10 |
| 3DSSG | 06.81 | 16.83 | 20.10 | 27.24 | 61.12 | 73.90 | **67.25** | **96.73** | **98.63** |
| SGFN | 06.70 | 15.59 | 18.65 | 35.51 | 67.38 | **79.35** | 48.62 | 92.32 | 97.70 |
| Ours | **07.84** | **18.69** | **21.55** | **37.22** | **67.78** | 79.05 | 36.87 | 90.13 | 96.87 |

(b) **3DSSG-l160**

Table 2: Scene graph prediction performance on previously unseen relationship triplets, objects, and predicates under geometric segmentation setting.

**Faster training convergence:** Figure 5a and 5d report the recalls for nodes (objects) and edges (relationships) during training of ESGNN and SGFN on both train and validation sets. The recall slope of ESGNN in the first 10 epochs (5000 steps) is significantly higher than that of SGFN. This shows that ESGNN has faster convergence and higher initial recall.

**Robustness against imbalanced data:** We additionally provide the training confusion matrices after the first 1000 steps in Figure 6. It is notably observable that ESGNN 6a, 6c shows clearer diagonal patterns compared to SGFN 6b, 6d, which was biased towards some dominant classes. These patterns demonstrate that ESGNN is more data-efficient than SGFN, as it does not need to generalize over rotations and translations of the data, while still harnessing the flexibility of GNNs in larger datasets. However, due to its high bias Satorras et al. (2021), our EGCL-based message passing performs well with limited data but struggles to learn the dataset's subtleties as the training size increases. Additionally, in later training steps, we observe sign of overfitting. We anticipate that this behavior aligns with the EGCL design that intentionally allows overfitting to fully fit the training graphs Satorras et al. (2021), in which its authors also conducted a thorough overfitting evaluation. Section 6.4 further discusses our experiments to mitigate this effect. Nevertheless, our model consistently outperforms the existing methods overall.

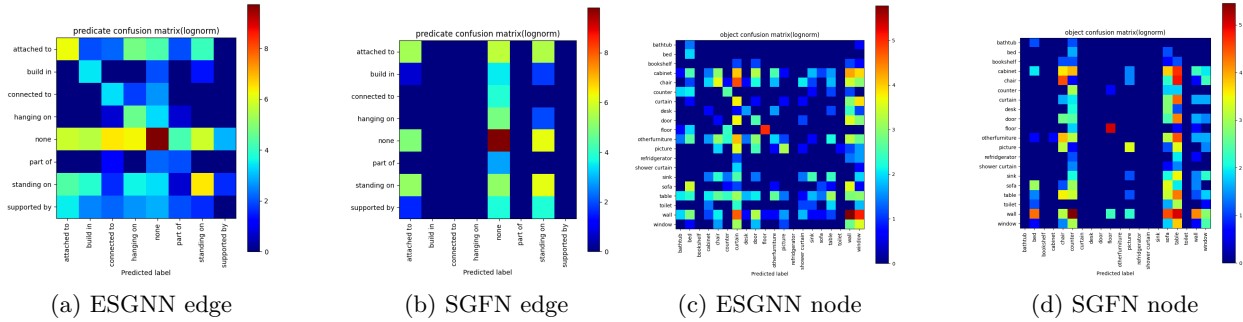

| (a) ESGNN edge | (b) SGFN edge | (c) ESGNN node | (d) SGFN node |

Figure 6: Confusion matrices after the first 1000 steps (2 epochs). ESGNN is better at handling imbalanced classes, as the predictions spread across multiple classes rather than "blindly" predicting towards dominant classes like SGFN.

**Computation Efficiency:** Table 3 reports the model parameters and runtime. In this benchmark, we run the inference for each method 1000 times and calculate the average runtime. All experiments were conducted on an AMD Threadripper Pro 3955WX 16-core CPU and an NVIDIA RTX 4090 24GB GPU. The results show that ESGNN achieves better computational efficiency than existing methods. On CPU, ESGNN requires 26% fewer parameters and achieves a 1.1× speedup over SGFN, with higher FPS. For broader context, OpenFusion Yamazaki et al. (2024) reports 50 FPS for 3D semantic map reconstruction on the ScanNet dataset. Although their setup differs from ours and hardware specifications were not reported, we provide this information as a useful reference for understanding the runtime efficiency of scene graph generation and VLM-based semantic map reconstruction.

| Method | Params | | CPU | | GPU | |
| --- | --- | --- | --- | --- | --- | --- |
| | GNN | Total | Runtime | FPS | Runtime | FPS |
| 3DSSG | 2,629,120 | 3,245,204 | 57.35 ms | 17.44 (10× ↓) | 2.51 ms | 398.00 (1.2× ↓) |
| SGFN | 2,245,312 | 2,861,396 | 6.27 ms | 159.48 (1.1× ↓) | 2.20 ms | 455.00 (1.1× ↓) |
| Ours | **1,780,577** | **2,396,661** | **5.61 ms** | **178.20** | **2.12 ms** | **471.78** |

Table 3: Performance comparison of different methods for scene graph generation, evaluated on both CPU and GPU. The reported runtime excludes the segmentation process and reflects only the scene graph generation phase, including node and edge prediction. ↓ denotes × times slower compared to ours.

**Scene Graph Extractor with Image Encoder:** ESGNN is also potentially applicable with a point-image encoder instead of 3D point clouds only. Figures 5c and 5f show the performance of our implementation with image encoders, called Joint-ESGNN, compared to existing methods, including JointSSG Wu et al. (2023) and SGFN.

## 6.4 Ablation Study

Table 4 evaluates ESGNN with different architectures and settings that we experimented with. ⓪ is the SGFN, run as the baseline model for comparison. ① is the ESGNN with 1 FAN-GCL layer and 1 EGCL layer. This is our best performer and is used for experiments in Section 6.3. ② is similar to ①. The only difference is that we concatenate the coordinate embedding to the output edge embedding after message passing. ③ and ④ are similar to ① and ②, respectively, but with 2 FAN-GCL layers and 2 EGCL layers.

| Method | Relationship | | Object | | Predicate | |
|---|---|---|---|---|---|---|
| | R@1 | R@3 | R@1 | R@3 | R@1 | R@2 |
| ⓪ SGFN | 37.82 | 48.74 | 62.82 | **88.08** | 81.41 | 98.22 |
| ① **ESGNN_1** | **42.30** | **53.30** | **63.21** | 86.70 | 94.34 | **98.30** |
| ② ESGNN_1X | 34.96 | 42.59 | 57.55 | 86.18 | 92.68 | 98.08 |
| ③ ESGNN_2 | 35.63 | 44.63 | 57.55 | 84.41 | 93.93 | 97.94 |
| ④ ESGNN_2X | 37.94 | 50.58 | 59.97 | 85.23 | **94.53** | 98.01 |

Table 4: Evaluation of different ESGNN architectures on the scene graph generation task using the 3DSSG-l20 dataset. ② is our best performer and is used for evaluation in Section 6.3.

Models ③ and ④ perform well on the train set but poorly on the validation and test sets, potentially suffering from overfitting as they contain more layers. Models ② and ④ have higher edge recalls in several initial epochs, but decline in the later epochs, as shown in Fig. 5b, 5e. To prevent overfitting, we notice that using a single EGCL layer is enough. We also experiment with different dropout rates between 0.3 and 0.5. However, a dropout rate of 0.5 led to underfitting, preventing the model from converging during training. We select dropout rate of 0.3 due to its balanced outcome.

## 6.5 Temporal Graph Matching Evaluation

**Experiment Setup:** We assess our work from two key perspectives. First, we evaluate our Graph Matching Network using different backbone scene graph extractors, including the state-of-the-art SGFN Wu et al. (2021b) and our proposed ESGNN. Second, we compare the effectiveness of the SGFN and ESGNN backbones on the state-of-the-art SG-PGM Xie et al. (2024). To ensure the same set-up, we reimplement and evaluate SG-PGM with our proposed dataset mentioned in Section 6.1. For top-K retrieval, we typically set $K \leq 5$ to focus on the most relevant nodes. Using a larger K typically tends to inflate the score, but adds little insight in evaluating the similarity matching.

| Method | Backbone | R@k | | | | MRR@k | |
|---|---|---|---|---|---|---|---|
| | | @1 | @2 | @3 | @5 | @2 | @3 |
| SG-PGM | SGFN | 10.61 | 19.16 | 28.56 | 44.71 | 14.88 | 18.02 |
| | **ESGNN** | **30.14** | **46.94** | **57.63** | **73.86** | **38.54** | **42.10** |
| SG-PGM (point) | SGFN | 18.54 | 33.28 | **45.73** | **62.82** | 25.91 | 30.06 |
| | **ESGNN** | **20.91** | **33.55** | 44.41 | 62.01 | **27.23** | **30.85** |
| Ours | SGFN | 54.48 | 72.95 | 82.11 | 90.45 | 63.71 | 66.77 |
| | **ESGNN** | **70.15** | **84.06** | **90.09** | **95.25** | **78.93** | **81.94** |

Table 5: Evaluation of the Graph Matching Network with 30 training epochs. Note that **R@1 = MRR@1** reflects node-matching accuracy.

**Better outcome with ESGNN backbone scene graph extractor:** Table 5 demonstrates that utilizing our proposed ESGNN as the backbone scene graph extractor consistently outperforms SGFN across all metrics, on both graph matching networks. This underscores the advantage of ESGNN's symmetry-preserving property, which enhances the representation of nodes and edges in 3D point clouds, making them more resilient to transformations and significantly improving graph matching performance.

**Better outcome with our Graph Matching Network:** Although SG-PGM Xie et al. (2024) achieved competitive results for node matching in their setting, it performs poorly in our scenario, where node and edge features are extracted from a scene graph extractor model rather than one-hot encoded from the ground truth. In our case, both nodes and edges have more labels, leading to several ambiguous predictions, including overlaps, incorrect assignments, and permutations. This explains why SG-PGM fails in this setting. In contrast, our model leverages the equivariant properties of nodes and edges while applying sum pooling, ensuring permutation invariance and significantly outperforming SG-PGM in this scenario, indicating the robustness of the model.

## 7    Conclusion

In this paper, we introduced the TESGNN, a novel method that overcomes key limitations in 3D scene understanding. By leveraging the symmetry-preserving property of the Equivariant GNN, our architecture ensures robust and efficient generation of semantic scene graphs from 3D point clouds. Our proposed Temporal Graph Matching Model provides a global representation from local scene graphs for real-time dynamic environments. Experimental results show that TESGNN outperforms state-of-the-art methods in accuracy, convergence speed, and computational efficiency.

**Limitations & Future Work** Future work will focus on optimizing TESGNN for complex real-world scenarios by integrating additional sensor modalities like LiDAR and RGB-D data to improve scene graph generation. We also aim to extend the model's temporal capabilities for continuous data streams, enhancing its suitability for autonomous navigation and multi-agent systems.

## A GitHub

We provide the [GitHub Repository](#) [3] for our TESGNN implementation.

## B Equivariance of ESGNN

This section demonstrates that our model is translation equivariant with respect to $\mathbf{x}$ for any translation vector $g \in \mathbb{R}^n$, and rotation and reflection equivariant with respect to $\mathbf{x}$ for any orthogonal matrix $Q \in \mathbb{R}^{n \times n}$. Formally, we prove that the model satisfies:

$$Q\mathbf{x}^{l+1} + g, \mathbf{h}^{l+1} = \text{ESGNN}\left(Q\mathbf{x}^l + g, \mathbf{h}^l\right)$$

### B.1 FAN-GCL Layer

**Node Update:**

$$\mathbf{h}_i^{\ell+1} = g_v\left(\left[\mathbf{h}_i^\ell, \max_{j \in \mathcal{N}(i)}\left(\text{FAN}\left(\mathbf{h}_i^\ell, \mathbf{e}_{ij}^\ell, \mathbf{h}_j^\ell\right)\right)\right]\right),$$

**Edge Update:**

$$\mathbf{e}_{ij}^{\ell+1} = g_e\left(\left[\mathbf{h}_i^\ell, \mathbf{e}_{ij}^\ell, \mathbf{h}_j^\ell\right]\right),$$

**Equivariance:** Assuming $\mathbf{h}^\ell$ is invariant to $\text{E}(n)$ transformations on $\mathbf{x}$—i.e., no information about the absolute position or orientation of $\mathbf{x}^\ell$ is encoded into $\mathbf{h}^\ell$—then the outputs $\mathbf{h}_i^{\ell+1}$ and $\mathbf{e}_{ij}^{\ell+1}$ of FAN-GCL are also invariant.

### B.2 EGCL Layer

**Node Update:**

$$h_i^{(l+1)} = h_i^{(l)} + g_v\left(\text{concat}\left(h_i^{(l)}, \sum_{j \in \mathcal{N}(i)} e_{ij}^{(l)}\right)\right),$$

**Edge Update:**

$$e_{ij}^{(l+1)} = g_e\left(\text{concat}\left(h_i^{(l)}, h_j^{(l)}, \|\mathbf{x}_i^{(l)} - \mathbf{x}_j^{(l)}\|^2, e_{ij}^{(l)}\right)\right),$$

**Coordinate Update:**

$$\mathbf{x}_i^{(l+1)} = \mathbf{x}_i^{(l)} + \sum_{j \in \mathcal{N}(i)} (\mathbf{x}_i^{(l)} - \mathbf{x}_j^{(l)}) \cdot \phi_{\text{coord}}(e_{ij}^{(l)}),$$

**Equivariance:** Since $\mathbf{h}^\ell$ is invariant to $\text{E}(n)$ transformations of $\mathbf{x}$, the output $e_{ij}^{(l+1)}$ of EGCL is also invariant because the distance between two particles is invariant to translations:

$$\left\|\mathbf{x}_i^l + g - \left[\mathbf{x}_j^l + g\right]\right\|^2 = \left\|\mathbf{x}_i^l - \mathbf{x}_j^l\right\|^2,$$

and invariant to rotations and reflections:

$$\begin{aligned}
\left\|Q\mathbf{x}_i^l - Q\mathbf{x}_j^l\right\|^2 &= \left(\mathbf{x}_i^l - \mathbf{x}_j^l\right)^\top Q^\top Q \left(\mathbf{x}_i^l - \mathbf{x}_j^l\right) \\
&= \left(\mathbf{x}_i^l - \mathbf{x}_j^l\right)^\top \mathbf{I} \left(\mathbf{x}_i^l - \mathbf{x}_j^l\right) \\
&= \left\|\mathbf{x}_i^l - \mathbf{x}_j^l\right\|^2
\end{aligned}$$

---

[3]Link: https://github.com/HySonLab/TESGraph/tree/main

Thus, the edge operation becomes invariant:

$$e_{ij}^{(l+1)} = g_e \left( \mathbf{h}_i^l, \mathbf{h}_j^l, \left\| Q\mathbf{x}_i^l + g - \left[ Q\mathbf{x}_j^l + g \right] \right\|^2, e_{ij}^{(l)} \right)$$
$$= g_e \left( \mathbf{h}_i^l, \mathbf{h}_j^l, \left\| \mathbf{x}_i^l - \mathbf{x}_j^l \right\|^2, e_{ij}^{(l)} \right)$$

The coordinates $\mathbf{x}$ are also $E(n)$ equivariant. We prove this by showing that an $E(n)$ transformation of the input yields the same transformation of the output. Since $e_{ij}^{(l+1)}$ is already invariant, we aim to show:

$$Q\mathbf{x}_i^{l+1} + g$$
$$= Q\mathbf{x}_i^l + g + C \sum_{j \neq i} \left( Q\mathbf{x}_i^l + g - \left[ Q\mathbf{x}_j^l + g \right] \right) g_{coord} \left( e_{ij}^{(l+1)} \right)$$

We have the following derivation:

$$Q\mathbf{x}_i^l + g + C \sum_{j \neq i} \left( Q\mathbf{x}_i^l + g - Q\mathbf{x}_j^l - g \right) g_{coord} \left( e_{ij}^{(l+1)} \right)$$
$$= Q\mathbf{x}_i^l + g + QC \sum_{j \neq i} \left( \mathbf{x}_i^l - \mathbf{x}_j^l \right) g_{coord} \left( e_{ij}^{(l+1)} \right)$$
$$= Q \left( \mathbf{x}_i^l + C \sum_{j \neq i} \left( \mathbf{x}_i^l - \mathbf{x}_j^l \right) g_{coord} \left( e_{ij}^{(l+1)} \right) \right) + g$$
$$= Q\mathbf{x}_i^{l+1} + g$$

Thus, a transformation $Q\mathbf{x}^l + g$ on $\mathbf{x}^l$ leads to the same transformation on $\mathbf{x}^{l+1}$, while $\mathbf{h}^{l+1}$ remains invariant, satisfying:

$$Q\mathbf{x}^{l+1} + g, \mathbf{h}^{l+1} = \text{EGCL} \left( Q\mathbf{x}^l + g, \mathbf{h}^l \right).$$

Both FAN-GCL and EGCL layers are designed to maintain equivariance under translation, rotation, and permutation transformations, making them well-suited for tasks requiring spatial and relational reasoning in graph-based neural networks.

## C  Towards Temporal Model's Training and Future Works

### C.1  Loss Selection and Training With Multiple Losses

This section explores various training strategies and loss functions applicable to our Temporal Similarity Matching. Our Temporal Model adopts the training objectives of embedding models, which are to optimize the embedding model such that distances between similar embeddings are minimized, while those between dissimilar embeddings are maximized. There are several common training strategies [4] Reimers & Gurevych (2019) as mentioned below.

**Contrastive Loss:**  This loss expects an embedding pair and a label of either 0 or 1. For label 1, the distance between the two embeddings is reduced. Otherwise, the distance between the embeddings is increased.

**Online Contrastive Loss:**  This leverages the Contrastive Loss, but computes the loss only for pairs of hard positive (positives that are far apart) or hard negative (negatives that are close) per iteration. This loss empirically produces better performances. In our training setting for the Temporal Model, we utilize this as our main loss.

---

[4]Sentence Transformer Docs: We mostly refer to this well-scripted documentation for loss selection as well as different training strategies. Although its main application is for text embedding, it is very helpful to our proposed approach.

**Multiple Negatives Ranking Loss:** This loss expects a batch consisting of embedding pairs $(a_1, p_1), (a_2, p_2)..., (a_n, p_n)$ where we assume that $(a_i, p_i)$ are a positive pair and $(a_i, p_j), i \neq j$ are negative pairs. It then minimizes the negative log-likelihood for softmax normalized scores. To improve the robustness of the model, we can also provide multiple hard negative pairs per batch during training.

**Cosine Similarity Loss:** A fundamental approach. In this loss, the similarity of embeddings is computed with Cosine Similarity and compared (by Mean-Squared Error) to the labeled similarity score. In our case, we can treat label 0 (dissimilar pairs) and label 1 (similar pairs) like a float similarity score. However, this loss often takes a significantly longer time to compute due to the cosine similarity calculation.

**Training strategy with multiple losses:** In this strategy, we calculate each loss and update the model weights iteratively per training iteration. An example setup is to calculate the Contrastive Loss followed by the Multiple Negatives Ranking Loss. This strategy often enhances the outcome model and provides better results. In future work, we will improve our Temporal Model by applying this strategy.

### C.2 Extended Discussion on Point Cloud Data Preprocessing

As our approach is built upon segmentation of 3D point cloud data, we assume that the segmentation results are reasonably accurate. Nonetheless, it is important to emphasize that the preprocessing stage in segmentation should not be overlooked. Recent studies, such as Nguyen et al. (2024b), propose lightweight methods to refine segmentation inaccuracies through non-parametric statistical approaches. Meanwhile, Grčić et al. (2023) employs CNN-based architectures that encode RGB and depth images separately and utilize their extracted features to correct errors in RGB-based segmentation. We suggest that such denoising strategies can substantially improve segmentation quality, thereby positively influencing the overall performance of scene graph generation.

## D Implementation Details for ESGNN

### D.1 Network Architecture

The setup for the ESGNN architecture is provided in the code block below. Our output node and edge dimensions are set to be 256. As discussed in Section 6.3. Ablation Study of the main paper, we experimented with different layer settings for the training. Each ESGNN layer is defined as a combination of 1 FAN-GCL layer followed by 1 EGCL layer to produce the final node and edge embeddings. For the FAN-GCL, we use 8 Attention heads.

```
model:
    node_feature_dim: 256
    edge_feature_dim: 256
    gnn:
        hidden_dim: 256
        num_layers: 1
        num_heads: 8
        drop_out: 0.3
```

### D.2 Training Details

We adopt a similar training strategy to state-of-the-art works 3DSSG Wald et al. (2020) and Scene Graph Fusion Wu et al. (2021b). This ensures the results of our method ESGNN, are comparable to these existing results. Our params setup is as follows:

Following Wu et al. (2021b), we leverage the AdamW optimizer with Amsgrad and an adaptive learning rate inversely proportional to the logarithm of the number of edges. Given a training batch containing $n$ edges

```
training:
    max_epoch: 200
    lr: 1e-4
    patient: 30
    optimizer: 'adamw'
    amsgrad: true

    lambda_mode: dynamic # calculate the ratio of the number of node and edge.
    lambda_node: 0.1 # learning rate ratio
    lambda_edge: 1.0 # learning rate ratio

    scheduler:
        method: reduceluronplateau
        args: {gamma: 0.5,
               factor: 0.9}

    model_selection_metric: iou_node_cls
    # Select and save best model based on the IoU
    model_selection_mode: maximize
    # can be maximize or minimize.
    # e.g. minimize if "loss", maximize if "accuracy" or "IOU"

    metric_smoothing:
        method: ema

eval:
    topK: 10 # Evaluate with topk = 1, 3, 5, 10, etc.
```

Figure 7: Training configuration for ESGNN.

and a base learning rate $\text{lr}_{\text{base}} = 1 \times 10^{-3}$, the learning rate is adjusted as follows:

$$\text{lr} = \text{lr}_{\text{base}} \cdot \frac{1}{\ln n}.$$

We set the maximum training epoch to 200, with a learning rate of 0.0001.

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
