# OpenReview forum: "TESGNN: Temporal Equivariant Scene Graph Neural Networks for Efficient and Robust Multi-View 3D Scene Understanding"
_TMLR — Accepted by TMLR_

### Review · Reviewer_S7eR · 2025-08-19

**Summary Of Contributions:**

This paper introduces temporal equivariant scene graph neural networks (TESGNN). It takes as input a sequence of point clouds and generate scene graphs robust to translation and rotation changes by using an equivariant graph convolution layer (EGCL). Matching the different graphs across time, the approach generates a global scene representation.

*Strengths*

- The proposed approach can help identify relationships in scenes with higher recall.
- The equivariance property allows better global scene graph generation across time.

*Weaknesses*

- Technical novelty for the ESGNN scene graph extractor is limited.

**Audience:**

Yes

**Audience Explanation:**

The paper presents an application of equivariant graph neural networks, which are of interest to some of the TMLR audience, to scene graph understanding. This equivariant property allows modeling the scene across time or different viewpoints.

**Claims And Evidence:**

Yes

**Claims Explanation:**

Mostly yes, with some caveats.

"higher accuracy" could be defined with more nuance. In table 1 on page 9, the proposed method indeed achieves better relationship recall compared to the 2 baselines. However, the proposed approach is not always superior on all other metrics. I appreciate the authors including results on unseen data and acknowledging the similar results between their approach and SGFN.

I would also like to better understand if recall is sufficient as a metric.

Convergence is indeed faster compared to SGFN based on figure 5a and 5d, although I am wondering whether some hyper-parameter changes could affect this.

While temporal graph matching is somewhat possible with other scene graph generation methods, equivariance leads to clearly better results.

The proposed temporal matching method also handles non-ground truth inputs better compared to the SG-PGM baseline.

**Requested Changes:**

*Critical*

- On page 8, can you clarify why recall should be sufficient? If the model generates additional nodes or edges, should that be penalized?

- Are there more recent baselines?

*Would strengthen work*

- At the bottom of page 6, can you provide an intuitive explanation of the formulas?

- In the footnote on page 8, what are the issues with the "l160" dataset?

- On page 11, why 30 epochs?

*Typos or other*

p3. "Scence"

p8. The link in the footnote currently includes the ".", so it doesn't work directly.

---

> ### Author Response · Authors · 2025-09-21
> **Our rebuttal and comments**
>
> Thank you for your feedback. For typo-related errors, we will fix them in the revision. We provide our rebuttal to the reviewer's questions below.
>
> **On page 8, can you clarify why recall should be sufficient? If the model generates additional nodes or edges, should that be penalized?**
>
> We adopted the settings from the existing baselines [Lu, et al., 2016, Wu, et al., 2021, Wu, et al., 2023]  for better comparison of the performances. In this case, Mean Average Precision (mAP) is another widely used metric. However, mAP is a pessimistic evaluation metric because we can not exhaustively annotate all possible relationships. [Lu, et al., 2016]
>
> An example from [Lu, et al., 2016]: Consider the case where a model predicts <person - taller than - person> Even if the prediction is correct, mAP would penalize the prediction if we do not have that particular ground truth annotation.
>
> References:
> * Lu, C., et al. (2016, September). Visual relationship detection with language priors. ECCV (pp. 852-869). Cham: Springer International Publishing.
> * Wu, S. C., et al. (2021). Scenegraphfusion: Incremental 3d scene graph prediction from RGB-D sequences. IEEE/CVF CVPR (pp. 7515-7525).
> * Wu, S. C., Tateno, K., Navab, N., & Tombari, F. (2023). Incremental 3d semantic scene graph prediction from rgb sequences. IEEE/CVF CVPR (pp. 5064-5074).
>
> **Are there more recent baselines?**
>
> We noticed that some recent works used 3DSSG dataset for evaluation. However, most recent papers [Feng et al., 2025; Koch et al., 2024a,c] focus on **image or language embeddings** (often derived from vision–language encoders). In contrast, our work targets a different problem: investigating **how equivariance can improve 3D point cloud understanding**, leading to better generalization and faster convergence. Our focus is thus on the 3D backbone design rather than on multimodal embeddings.
>
> Besides, they did not use a similar setting to our paper. For example, [Koch et al. (2024a; 2024b; 2024c)] are using another dataset (ScanNet) to pretrain the node feature, which results in a better performance. Meanwhile, we demonstrate that TESGNN can still achieve a high result from the original without any pre-training.
>
> References:
> * Feng, M., et al. (2025). History-enhanced 3D scene graph reasoning from RGB-D sequences. IEEE TCSVT.
> * Koch, S., et al. (2024a). Sgrec3d: Self-supervised 3D scene graph learning via object-level scene reconstruction. WACV.
> * Koch, S., et al. (2024b). Lang3dsg: Language-based contrastive pre-training for 3D scene graph prediction. 3DV.
> * Feng, M., et al. (2025b). Hyperrectangle embedding for debiased 3D scene graph prediction from RGB sequences. IEEE TPAMI.
> * Koch, S., et al. (2024c). Open3dsg: Open-vocabulary 3D scene graphs from point clouds with queryable objects and open-set relationships. CVPR.
>
> **At the bottom of page 6, can you provide an intuitive explanation of the formulas?**
>
> The formulas explain how TESGNN updates the message passing to ensure the equivariance of the model. For the original backbone, the message passing is invariant because they only use the invariant features, such as “distance from 2 objects”. Here we want to include the spatial information in the feature (here is the 3D coordinate of the bounding-boxes), but still maintaining the equivariance of the model. The proof is given in **Appendix C**.
>
> **In the footnote on page 8, what are the issues with the "l160" dataset?**
>
> For the I160 dataset, we encountered an error when trying to implement the dataset, as we couldn’t load some objects and samples in the dataset. We also emailed the author of the dataset and discussed with them how to fix the issues, but we couldn’t fix it at that time.  However, after implementing the dataset on a new computer, we can run it now. We provide an evaluation table of TESGNN on the I160 dataset in the rebuttal comment for the reviewer f534.
>
> **On page 11, why 30 epochs?**
>
> For running on l20 dataset, 30 epochs are already sufficient for training. We also monitor the loss on test sets and terminate the training if overfitting occurs as the test loss increases.

---

### Review · Reviewer_s3KQ · 2025-08-28

**Summary Of Contributions:**

This work studies the multi-view 3D scene understanding. The authors propose a new GNN-based architecture called TESGNN, consisting of an equivariant scene GNN that extracts scene graphs from 3D point clouds with symmetry preserved, and a temporal graph matching process based on the embeddings produced from the equivariant GNN. The authors evaluate TESGNN on multiple benchmarks that demonstrate the empirical improvements of TESGNN.

**Audience:**

Yes

**Audience Explanation:**

As the studied problem is quite important to a number of practical applications, if the claims were well-supported, the results are interesting and valuable to the community.

**Claims And Evidence:**

No

**Claims Explanation:**

Since this paper does not provide any theoretical results, the claims need to be supported by empirical evaluations. However, the existing evaluations may not be sufficient:
- From the tables, it can be found that TESGNN still underperforms previous methods in some cases;
- When the image embeddings of the scenes are incorporated, the differences between TESGNN and SGFN become smaller. It seems to imply that the symmetries do not matter too much for this task;
- Regarding "More robust against imbalanced data", it's unclear how "clearer diagonal patterns" and "biases towards dominant classes" are related to the robustness induced by symmetry preserving, and how the two observations imply the data efficiency;

**Requested Changes:**

1. How the symmetries are preserved through EGCL could be better explained.

2. The matching process can be better presented in the form of algorithms.

3. Some experimental details are not clear:
- How are the hyperparameters selected?
- Is there a validation set for hyperparameter selection?
- There is no report on the variance or error bars in the evaluation.

Minor:
- The superscripts in Figure 3 were not undefined before.

---

> ### Author Response · Authors · 2025-09-21
> **Our rebuttal and comments**
>
> Thank you for your feedback. For typo-related errors, we will fix them in the revision. We provide our rebuttal to the reviewer's questions below.
>
> **Since this paper does not provide any theoretical results, the claims need to be supported by empirical evaluations. However, the existing evaluations may not be sufficient:**
>
> We thank the reviewer for this comment. We would like to clarify that our work **does provide a theoretical proof of equivariance**, in **Appendix C**. This addresses the concern that our claims lack theoretical grounding.
>
> Regarding empirical support, it is well-established in the literature that equivariant architectures achieve **faster convergence** and greater **data efficiency** in 3D domains. For example, NequIP demonstrates state-of-the-art accuracy with a fraction of the data required by non-equivariant baselines [Batzner et al., 2022]. Tensor Field Networks [Thomas et al., 2018] and SE(3)-Transformers [Fuchs et al., 2020] show that equivariance eliminates the need for extensive data augmentation and accelerates training. Similarly, E(n)-GNNs [Satorras et al., 2021] and Cormorant [Anderson et al., 2019] report strong sample efficiency and robustness in molecular and geometric learning tasks. These works consistently validate the claim that equivariance reduces data requirements and improves convergence.
>
> To strengthen clarity, we will revise the manuscript to highlight our theoretical proof earlier in the main text and cite these prior results to better contextualize our empirical findings.
>
> References:
>
> * Batzner, S., et al. (2022). E(3)-equivariant graph neural networks for data-efficient and accurate interatomic potentials. Nature Communications, 13(1), 2453.
> * Thomas, N., et al. (2018). Tensor field networks: Rotation- and translation-equivariant neural networks for 3D point clouds. arXiv:1802.08219.
> * Fuchs, F. B., et al. (2020). SE(3)-transformers: 3D roto-translation equivariant attention networks. NeurIPS 2020.
> * Satorras, et al. (2021). E(n) equivariant graph neural networks. ICML 2022.
> * Anderson, B., Hy, T. S., et al. (2019). Cormorant: Covariant molecular neural networks. NeurIPS 2019.
>
> **From the tables, it can be found that TESGNN still underperforms previous methods in some cases.**
>
> There are still a few metrics that our model is slightly lower. But for the overall triplet metric (including objects, predicates, and relationships), our method is clearly better than previous methods. Moreover, our method focuses more on how fast the model can converge within very few steps of training.
>
> **When the image embeddings of the scenes are incorporated, the differences between TESGNN and SGFN become smaller. It seems to imply that the symmetries do not matter too much for this task.**
>
> As the image gives much more information compared to the 3D point clouds, the effect is a bit smaller compared to the setting of using only 3D point clouds. However, in Figure (5f), we demonstrated that using our ESGNN backbone can help improve the model’s performance in classifying the object along all the training steps.
>
> **Regarding "More robust against imbalanced data", it's unclear how "clearer diagonal patterns" and "biases towards dominant classes" are related to the robustness induced by symmetry preserving, and how the two observations imply the data efficiency;**
>
> The diagonal patterns in the confusion matrix represent cases where predictions align with the ground-truth labels. Under imbalanced training data, the models often display vertical patterns corresponding to dominant classes, as predictions are skewed toward these majority classes. These vertical patterns imply bias and reduce generalization to minority classes. As shown, TESGNN exhibited better prediction towards the true labels **within the first few training steps**, which showed a clearer diagonal pattern. This indicates that the model’s correct predictions are spread among all classes, not only the dominant one.
>
> **How the symmetries are preserved through EGCL could be better explained.**
>
> Please refer to **Appendix C** for the proof of the symmetry preservation of TESGNN.
>
> **The matching process can be better presented in the form of algorithms.**
>
> We will change the script in the revision.
>
> **How are the hyperparameters selected? Is there a validation set for hyperparameter selection?**
>
> As reported in section 6.1, we have a validation set to test different settings, including an ablation study, and monitor the checkpoints’ results throughout training epochs. For most hyperparameters, we aligned our settings with prior works such as 3DSSG and SGFN. We provide information and explanation in **Appendix E.2**. In our experiments, we tried different dropout values,  from 0.3 to 0.5. However, a 0.5 dropout value resulted in underfitting, in which the model couldn’t converge during training. We selected the setting that yielded the best validation performance.

---

> > ### Comment · Reviewer_s3KQ · 2025-10-07
> >
> > Thank you for the detailed explanation to my concerns! I believe incorporating the response above could strengthen the clarity and make the contributions and results more accessible to the readers.

---

### Review · Reviewer_f534 · 2025-09-07

**Summary Of Contributions:**

This paper introduces TESGNN (Temporal Equivariant Scene Graph Neural Networks) for multi-view 3D scene understanding, which is a novel architecture that combines the following key innovations:

1. An Equivariant Scene Graph Neural Network (ESGNN) that leverages equivariant graph convolution and feature-wise attention to generate scene graphs from point clouds while preserving rotational and translational symmetries.

2. A Temporal Graph Matching Network that aligns and merges local scene graphs across time by embedding relational triplets (<subject, predicate, object>) and training with contrastive loss.

The authors evaluate on the 3DSSG dataset, showing that TESGNN improves recall, convergence speed, and robustness over prior approaches (3DSSG baseline and SGFN).

**Audience:**

Yes

**Audience Explanation:**

TESGNN presents a novel combination of equivariant GNNs and temporal matching for 3D scene understanding, both of which are active areas of research. The work will be of interest to some TMLR readers working on geometric deep learning, robotics perception, and temporal graph learning. At the same time, the impact and accessibility could be improved by testing on more datasets, adding clearer intuition for the temporal model, and expanding the ablation studies.

**Broader Impact Concerns:**

This paper is primarily methodological and technical. The societal impact stems from applications in robotics and autonomous systems. While no immediate risks are apparent, it would be appropriate to briefly note that improved scene understanding can be used for both positive applications (e.g., assistive robotics, navigation) and potentially concerning ones (e.g., surveillance).

**Claims And Evidence:**

Yes

**Claims Explanation:**

The paper’s claims are generally well supported by the experiments and ablations provided. I highlight some points below:

1. The claim that equivariance leads to more robust scene graphs is supported by stronger recall metrics and clearer diagonals in the confusion matrices (Fig. 6).

2. The claim of faster convergence is substantiated by training curves showing higher recall within the first few epochs compared to SGFN (Fig. 5).

3. The claim that temporal matching yields more coherent global graphs is borne out by the superior performance of the proposed temporal model relative to SG-PGM in Table 4.

4. The ablation studies provide some evidence for the design choices, though they focus mainly on layer depth rather than fully disentangling all contributions.

5. The paper is careful in acknowledging limitations (overfitting tendencies, dataset scope), which adds credibility to the claims that are made.

**Requested Changes:**

I give explicit requested changes below, noting which are critical for securing acceptance:

1. Please evaluate TESGNN on at least one dataset beyond 3DSSG-l20, even at smaller scale. This will greatly help demonstrate generality. [Critical for securing acceptance]

2. Since the paper notes overfitting issues with ESGNN (Sec. 6.3), provide a more detailed analysis and discuss possible mitigations (e.g., dropout, augmentation, regularization). [Critical for securing acceptance]

3. I would like the authors to make some clarifications in the paper [important, but not critical for securing acceptance]:

- Expand Sec. 5 with a more intuitive explanation or example of how triplet embeddings are constructed and matched across time

- Report standard deviations (across random seeds) for recall metrics to make improvements more interpretable in terms of statistical significance

- State explicitly how thresholds for top-K retrieval and similarity cutoffs were chosen, and provide a brief sensitivity analysis

4. I also have some minor requested changes in terms of writing [recommended, but not critical for securing acceptance]:

- Ensure parenthetical citation (\citep) is used where appropriate instead of \citet; this formatting mistake occurs in several place (e.g. top of page 6)

---

> ### Author Response · Authors · 2025-09-21
> **Our rebuttal and comments**
>
> Thank you for your feedback. For typo-related errors, we will fix them in the revision. We provide our rebuttal to the reviewer's questions below.
>
> **Please evaluate TESGNN on at least one dataset beyond 3DSSG-l20**
>
> We appreciate the reviewer’s suggestion. In addition to the 3DSSG-l20 dataset, we have now evaluated TESGNN on the larger **3DSSG-l160 dataset**, and the results are reported in the table below. We will add these results in the revision.
>
> Model   | Objects | Predicates | Triplets R@10
> --------|---------------|-----------------|---------------
> SGFN    | 0.358         | 0.141           | 0.694
> 3DSSG | 0.273         | **0.216**       | 0.699
> ESGNN   | **0.375**     | 0.209           | **0.705**
>
> Model   | Task       | R@1   | R@5   | R@10
> --------|------------|-------|-------|-------
> SGFN | Triplets   | 0.647 | 0.682 | 0.694
> SGFN | Objects    | 0.358 | 0.675 | **0.794**
> SGFN | Predicates | 0.486 | 0.923 | 0.977
> 3DSSG | Triplets   | 0.648 | 0.687 | 0.699
> 3DSSG | Objects    | 0.273 | 0.608 | 0.736
> 3DSSG | Predicates | **0.672** | **0.967** | **0.986**
> ESGNN   | Triplets   | **0.650** | **0.694** | **0.705**
> ESGNN   | Objects    | **0.375** | **0.680** | 0.791
> ESGNN   | Predicates | 0.369 | 0.901 | 0.969
>
>
> Model   | Task     | New R@1  | New R@5  | New R@10
> --------|------------|-------|-------|-------
> SGFN    | Triplets   | 0.067 | 0.156 | 0.187
> SGFN    | Objects    | 0.355 | 0.674 | **0.793**
> SGFN    | Predicates | 0.486 | 0.923 | 0.977
> 3DSSG | Triplets   | 0.068 | 0.168 | 0.201
> 3DSSG | Objects    | 0.272 | 0.611 | 0.739
> 3DSSG | Predicates | **0.672** | **0.967** | **0.986**
> ESGNN   | Triplets   | **0.078** | **0.187** | **0.216**
> ESGNN   | Objects    | **0.372** | **0.678** | 0.790
> ESGNN   | Predicates | 0.369 | 0.901 | 0.969
>
> To the best of our knowledge, **3DSSG is the only available benchmark that provides semantic scene graphs**. Other 3D scene graph datasets focus on hierarchical graphs, which primarily capture room-object membership rather than semantic relationships between objects. This distinction is noted in several recent works [Feng et al., 2025a,b, Koch et al., 2024a,b, c]
>
> It is also important to note that recent papers [Feng et al., 2025; Koch et al., 2024a,c] focus on **image or language embeddings** (often derived from vision–language encoders). In contrast, our work targets a different problem: investigating **how equivariance can improve 3D point cloud understanding**, leading to better generalization and faster convergence. Our focus is thus on the 3D backbone design rather than on multimodal embeddings.
>
> Besides, they did not use a similar setting to our paper. For example, Koch et al. (2024a; 2024b; 2024c) are using another dataset (ScanNet) to pretrain the node feature, which results in a better performance. Meanwhile, we demonstrate that TESGNN can still achieve a high result from the original without any pre-training.
>
> References:
> * Feng, M., et al. (2025). History-enhanced 3D scene graph reasoning from RGB-D sequences. IEEE TCSVT.
> * Koch, S., et al. (2024a). Sgrec3d: Self-supervised 3D scene graph learning via object-level scene reconstruction. WACV.
> * Koch, S., et al. (2024b). Lang3dsg: Language-based contrastive pre-training for 3D scene graph prediction. 3DV.
> * Feng, M., et al. (2025b). Hyperrectangle embedding for debiased 3D scene graph prediction from RGB sequences. IEEE TPAMI.
> * Koch, S., et al. (2024c). Open3dsg: Open-vocabulary 3D scene graphs from point clouds with queryable objects and open-set relationships. CVPR.
>
> **For overfitting issues with ESGNN (Sec. 6.3), provides a more detailed analysis and discusses possible mitigations (e.g., dropout, augmentation, and regularization).**
>
> We observed that our model suffered from overfitting when the number of training epochs was large. One solution is to reduce the number of training epochs and monitor the checkpoints’ results on the validation set and test set. We also tried different dropout values, ranging from 0.3 to 0.5. However, a 0.5 dropout value resulted in underfitting, in which the model couldn’t converge during training. We selected the best result from trying different settings.
>
> Another effort we made is provided in the Ablation study section. Through this experiment, we noticed that a single EGCL layer is enough to prevent overfitting. Using 2 or more EGCL made the training performance increase, but the test results decreased. This comes from the EGCL designs to fit well within a substantial amount of data during training, which its authors also mentioned and conducted a thorough overfitting evaluation (Satorras et al., 2021).
>
> Reference:
> * Satorras, V.G., et al. (2021). E(n) equivariant graph neural networks. ICML 2022.
>
> **Expand Sec. 5 with a more intuitive explanation of how triplet embeddings are constructed and matched across time. Report standard deviations for recall metrics to make improvements more interpretable in terms of statistical significance**
>
> We will add this in the revision.

---

> > ### Author Response · Authors · 2025-09-22
> > **Additional comments**
> >
> > We provide some additional comments due to the letter limit of our previous comment.
> >
> > **For overfitting issues with ESGNN (Sec. 6.3), provides a more detailed analysis and discusses possible mitigations**
> >
> > We mentioned our answer in the above comment. We will highlight these points and discuss them in the revision for this paper.
> >
> > **State explicitly how thresholds for top-K retrieval and similarity cutoffs were chosen, and provide a brief sensitivity analysis**
> >
> > For top-K retrieval, we typically set $K <= 5$ to focus on the most relevant nodes. Using a larger $K$ typically tends to inflate the score (approaching 1.0), but adds little insight in evaluating the similarity matching. In our revision, we will include an analysis by plotting how the performance varies across different cut-off thresholds and $K$ values.

---

> > > ### Comment · Reviewer_f534 · 2025-10-03
> > > **Thank you for the rebuttal**
> > >
> > > Thank you for the additional information provided in your rebuttal! I request that this information be included in the updated version of the paper, but otherwise, I believe my concerns have been mostly assuaged.

---

### Review · Reviewer_zS6Y · 2025-10-07

**Summary Of Contributions:**

The paper outlines an equivariant scene graph model that mixes attention with E(n) equivariant graph convolutions to extract 3d scene graphs. The idea is that this approach is more robust to shifts in pose, which is well motivated. Additionally, they add a temporal graph matching module that looks to fuse local scene graphs across sequences via triplet embeddings and top-k retrieval.

### Strengths
1. The narrative is clear, and the authors apply a modular view. The triplet embeddings are both reasonable and sensible in the context of permutation aware operations in 3d scene graphs. Several innovations are novel and interesting for further work.
2. Strong results, faster convergence and better performance on average demonstrates clear benefits for the proposed framework.
3. Appendix provides a formal argument for equivariance, which states the conditions explicitly.
4. The authors clearly states the limitations inherent in the proposed approach.

### Weaknesses
1. Claims of efficiency gains would have benefited from more evidence, runtime / throughput results or parameter / FLOPs. The work demonstrates fast convergence, but this could be clarified in the claims.
2. "Outperforms SotA" is a go-to claim that may in aggregate be correct, but fails in some instances. This is typical for modern ML papers, and is not necessary as a blanket statement. The results speak for themselves, and the method successfully demonstrates benefits in several aspects. Slight modification of the claims would be prudent.
3. The role of reflection (SE(n) to E(n)) could be developed further. Handedness is arguably an important property for determining relations based on the perspective. This could be slightly elaborated on by the authors, and motivated in the text.

Overall, the paper is well formulated and demonstrates clear performance gains, while openly elaborating on the core argument and limitations. It introduces novel, intuitive ideas that can be beneficial in developing the field.

**Audience:**

Yes

**Audience Explanation:**

The contributions of the paper are novel, improving existing methods with fresh ideas and methodology that can be applied to drive the field forward. While somewhat niche, research on scene graphs for 3d understanding and reasoning is promising, particularly in applied settings. In this reviewers opinion, the work is relevant and interesting.

**Broader Impact Concerns:**

As the paper is largely technical, this reviewer sees no concerns that needs to be flagged for ethical or societal implications.

**Claims And Evidence:**

No

**Claims Explanation:**

The core issues this reviewer has with the current draft are regarding some slightly overstated claims, as mentioned in the previous section. On the whole, the authors are being very careful about how they frame their method and its limitations, which this reviewer appreciates.

However, this reviewer believes the claims on performance and efficiency should be clarified with slightly more precision than the current draft admits. Simply clarifying that "computational efficiency" is an empirical result on convergence, and rephrasing state-of-the-art to a milder tone would go a long way in informing the reader about the nuances in the results. In particular, for practical applications (e.g., robotics) the claims of efficiency relate more to real-time performance, hence demonstrations of throughput is arguably more informative than convergence results.

The reviewer emphasises that these should be relatively simple to incorporate in the final draft without detracting from the main contributions of the paper.

**Requested Changes:**

1. Is reflection equivariance a desirable property for 3d scene graphs? Can the authors elaborate on why this is not included as a discussion in terms of the problem itself? Arguably, SE(n) would suffice if handedness is to be preserved. This would strengthen the work.
2. Clarify the nuances in claims of state-of-the-art results, and have the claims of the paper accurately depict the results. This is critical for acceptance. In particular, the abstract and contributions should be slightly revised to match the results.
3. Similarly, clarify the nuances of what is meant by computational efficiency. If the authors want to retain the current claims, they should include empirical results on throughput or runtime, or theoretical benefits in terms of FLOPs. Adjusting the claim or including these results to fit the current claims is critical for this reviewer to recommend the paper to be accepted.

---

> ### Author Response · Authors · 2025-10-20
> **Our rebuttal and comments**
>
> Thank you for your feedback. We provide our answers to the reviewer's questions below.
>
> **Is reflection equivariance a desirable property for 3d scene graphs? Can the authors elaborate on why this is not included as a discussion in terms of the problem itself? Arguably, SE(n) would suffice if handedness is to be preserved. This would strengthen the work.**
>
> We appreciate the reviewer’s insightful comment. We agree that reflection equivariance is not required in the standard setup for 3D scene graph generation, and that SE(n) equivariance is sufficient for this setup. The main objective of our work is to highlight that incorporating equivariant GNNs can be beneficial for scene graph generation and offer an efficient alternative to VLM/ image-encoder approaches.
>
> In our implementation, we chose E(n) GNN primarily due to its popularity and easy implementation. However, in our code, the backbone equivariant GNNs can be replaced with other equivariant architectures, such as SO(3). We do hope this work encourages further exploration of different forms of equivariance on scene graph generation.
>
> **Clarify the nuances in claims of state-of-the-art results, and have the claims of the paper accurately depict the results. This is critical for acceptance. In particular, the abstract and contributions should be slightly revised to match the results.**
>
> We appreciate the reviewer for the constructive feedback. We will revise the wording to fit the contribution of the paper.
>
> As the reviewer also points out, this paper aims to provide an alternative option for scene understanding and representation, instead of depending on heavyweight VLMs/LLMs. Our results show that leveraging Equivariant GNN is effective for scene graph generation compared to existing methods. We hope that through our experiment and addressing the limitations carefully, this work introduces the reader to an alternative solution that is effective and efficient for scene understanding.
>
> **Similarly, clarify the nuances of what is meant by computational efficiency. If the authors want to retain the current claims, they should include empirical results on throughput or runtime, or theoretical benefits in terms of FLOPs.**
>
> We want to clarify the “computational efficiency” by 2 major points:
> * Firstly, approaches that leverage graph representation (like ours) instead of semantic map representation enable efficient inference on the output scene graph. The output scene graphs can later be used for specific tasks such as navigation and path planning. We believe we elaborate this point quite clearly in the first part of the paper.
> * Secondly, we agree with the reviewer that the architecture of TESGNN should also be examined. This should affect the scene graph generation performance. In the revision for this paper, we will report the number of parameters and the runtime of TESGNN.

---

### Decision · Action_Editor_FFpp · 2025-10-07

**Recommendation:** Accept with minor revision

**Audience:**

Yes

**Audience Explanation:**

The work can draw interests from two seemingly disconnected subfields, geometric deep learning and 3D scene understanding.

**Claims And Evidence:**

Yes

**Claims Explanation:**

The work proposed an equivariant scene graph neural network for 3D scene understanding. The experiments in the paper and the rebuttal are fairly strong, though it focuses on only one dataset.

---

> ### Author Response · Authors · 2025-10-29
> **Response by authors**
>
> We appreciate your feedback and the input from the other reviewers. We’ve updated the camera-ready version to meet all reviewer requirements.